# SERBP1: A Multifunctional RNA-Binding Protein Linking Gene Expression, Cellular Metabolism, and Diseases

**DOI:** 10.3390/cells14211705

**Published:** 2025-10-30

**Authors:** Zezhao Ji, Abduxukur Ablimit

**Affiliations:** Department of Histology and Embryology, Basic Medical College, Xinjiang Medical University, Urumqi 830011, China; 18460308473@163.com

**Keywords:** SERBP1, RNA-binding protein, stress granules, tumor, reproductive regulation, nervous system

## Abstract

**Highlights:**

**What are the main findings?**
1.SERBP1, an intrinsically disordered RNA-binding protein with RG/RGG repeats, exhibits multifunctional roles in cellular physiology and pathology.2.It dynamically regulates stress granule assembly/clearance and ribosome dormancy/activation, enabling cells to adapt to environmental stresses.

**What are the implications of the main findings?**
3.SERBP1 participates in tumorigenesis, reproductive regulation, nervous system function, and viral infection through RNA metabolism and post-translational modification mechanisms.4.Its interactions with PARP1 and involvement in ribosome biogenesis and cell cycle regulation reveal novel molecular networks.

**Abstract:**

SERBP1 (SERPINE1 mRNA-Binding Protein 1), as an RNA-binding protein with multiple biological functions, has become a research hotspot in the field of life sciences in recent years. Its unique molecular structure, such as the presence of RG/RGG repeat sequences and the absence of typical RNA-binding domains, enables it to exert diverse roles in cells. This article systematically reviews the research progress of SERBP1 in various fields including cellular stress response, tumorigenesis and development, reproductive system regulation, nervous system function, and viral infection, elaborates on its mechanism of action in detail (including newly supplemented content on cell cycle regulation, interaction with PARP1, and ribosome biogenesis), and outlines future research directions. It aims to provide a reference for in-depth understanding of the biological functions of SERBP1 and the diagnosis and treatment of related diseases.

## 1. Introduction

SERBP1 was initially discovered for its ability to bind to SERPINE1 (Plasminogen Activator Inhibitor 1) mRNA and regulate its stability. Subsequent studies have gradually revealed its extensive roles in cellular physiological and pathological processes [1]. As an Intrinsically Disordered Protein (IDP), SERBP1 relies on its structural flexibility to interact with a variety of proteins and RNA molecules, participating in key cellular processes such as mRNA metabolism, translational regulation, and signal pathway transduction [2]. In recent years, with the continuous development of molecular biology techniques, the important roles of SERBP1 in stress response, tumorigenesis, reproductive development, neural function, and viral infection have been gradually clarified, and its functions in cell cycle regulation, ribosome biogenesis, and interaction with PARP1 have also been further explored, providing a new direction for the mechanism research of related diseases and the development of therapeutic targets [3,4,5].

As shown in Table 1, SERBP1 plays a core role in five major physiological and pathological processes including cellular stress response, tumorigenesis, reproductive regulation, nervous system function, and viral infection, achieving diverse biological effects by regulating different key cellular processes.

## 2. Molecular Structure and Post-Translational Modifications of SERBP1

### 2.1. Characteristics of Molecular Structure

The amino acid sequence of SERBP1 contains two key RG/RGG repeat domains, which serve as the important basis for its RNA-binding function and protein–protein interactions (Figure 1), although it lacks classical RNA-binding domains such as RNA Recognition Motif (RRM) and zinc finger structures [13]. Studies have found that the N-terminal and C-terminal regions of SERBP1 exhibit certain structural plasticity, and the C-terminal region also contains a stable α-helical structure, which may play an important role in its interaction with other molecules and subcellular localization [18]. In addition, as an intrinsically disordered protein, the disordered regions of SERBP1 endow it with the ability to bind to multiple ligands, enabling it to participate in the formation of dynamic molecular complexes such as Stress Granules (SGs) and nucleoli (membrane-less organelles), which is closely related to its functions in cellular stress response and RNA metabolism [5,19].

### 2.2. Post-Translational Modifications

The function of SERBP1 is finely regulated by a variety of post-translational modifications, among which arginine methylation, phosphorylation, and SUMOylation are relatively critical.

As summarized in Table 2, arginine methylation, phosphorylation, and other modifications of SERBP1 precisely regulate its function by changing subcellular localization.

#### 2.2.1. Arginine Methylation

Mediated by Protein Arginine Methyltransferase 1 (PRMT1), it mainly occurs in the RG/RGG domain of SERBP1. This modification not only affects the interaction of SERBP1 with RNA and other proteins but also plays an important regulatory role in its subcellular localization. For example, when SERBP1 is hypomethylated, it tends to localize in the nucleus/nucleolus; while in the normal methylated state, SERBP1 is mainly distributed in the cytoplasm and is more likely to be recruited to stress granules to participate in the stress response [20,21].

#### 2.2.2. Phosphorylation

Protein Kinase C ε (PKCε) can phosphorylate SERBP1. Phosphorylated SERBP1 can regulate the function of ribosomes, such as affecting its binding to the 40S ribosomal subunit, thereby regulating the process of protein translation [22]. During cell division, the phosphorylation state of SERBP1 may also be involved in regulating the cell cycle process to ensure the normal progression of cell division [26].

#### 2.2.3. SUMOylation and Ubiquitination

SERBP1 has multiple potential SUMOylation sites (lysine residues). SUMOylation can affect the interaction network of SERBP1 with other proteins, thereby regulating its functions in processes such as gene expression regulation and cellular stress response [23]. For example, SUMOylation may alter the role of SERBP1 in the formation of PML Nuclear Bodies (PML-NBs), participating in intracellular signal transduction and the maintenance of genome stability [27]. In addition, SUMOylation may also affect the role of SERBP1 in gene transcriptional regulation by changing its nucleocytoplasmic shuttling ability. Under oxidative stress conditions, SUMOylated SERBP1 is more likely to enter the nucleus, interact with transcription factors such as p53, and regulate the expression of apoptosis-related genes, thereby influencing the survival fate of cells [24]. Recent studies have also found that ubiquitination of SERBP1 may be involved in its protein degradation process. When cells are in a state of long-term stress, the ubiquitin-proteasome system can degrade SERBP1 to avoid the toxic damage to cells caused by excessive accumulation of stress granules [25].

### 2.3. Interaction with PARP1 and PARylation Regulation

SERBP1 can form a PARylation-dependent complex with PARP1 (Poly (ADP-ribose) Polymerase 1) (Table 3). This complex participates in the regulation of RNA splicing, cell division, and ribosome biogenesis. Specifically, PARP1-mediated PARylation modification can enhance the binding ability of SERBP1 to target RNA (such as pre-mRNA and rRNA), thereby promoting the efficiency of RNA splicing and rRNA processing. In cell division, the SERBP1-PARP1 complex is recruited to the spindle apparatus, regulating the separation of chromosomes by interacting with microtubule-related proteins. The absence of this complex leads to abnormal chromosome segregation and cell cycle arrest. Further studies have shown that the formation of the SERBP1-PARP1 complex is dependent on the PARylation activity of PARP1, and the inhibition of PARP1 activity can significantly reduce the interaction between SERBP1 and RNA [5].

## 3. The Role of SERBP1 in Cellular Stress Response

### 3.1. Assembly and Clearance of Stress Granules

Stress granules are membrane-less ribonucleoprotein complexes formed in the cytoplasm when cells respond to various stress conditions such as oxidative stress, heat stress, and viral infection (Figure 2). Their core function is to temporarily store untranslated mRNA, inhibit global protein translation, and at the same time protect mRNA from degradation, helping cells survive the stress period [6,30].

As a core component of stress granules, the role of SERBP1 in the dynamic regulation of stress granules has been confirmed by multiple studies. In the initial stage of stress, SERBP1 can directly interact with G3BP1 (Ras GTPase-Activating Protein-Binding Protein 1) through its RG/RGG domain, and at the same time recruit 26S proteasome-related proteins PSMD10 and PSMA3 to form a complex, providing structural support for the initial assembly of stress granules [6]. Immunofluorescence co-localization experiments show that SERBP1 and G3BP1 exhibit high co-localization in stress granules, and the deletion of SERBP1 leads to the inability of G3BP1 to aggregate effectively under stress conditions, significantly reducing the formation efficiency of stress granules [29].

In the stress recovery stage, the function of SERBP1 shifts from an “assembly promoter” to a “clearance regulator”. Studies have found that SERBP1 can interact with the E3 ubiquitin ligase RNF111 to promote K63-linked polyubiquitination of G3BP1. This non-degradative ubiquitination modification does not mediate the degradation of G3BP1 but acts as a molecular signal to recruit the 26S proteasome into the interior of stress granules, accelerating the disassembly of stress granules [6]. When SERBP1 is knocked out, the catalytic activity of the 20S proteasome in cells decreases by approximately 40%, and the localization of ubiquitination-related proteins VCP (Valosin-Containing Protein) and FAF2 (UBX Domain-Containing Protein 2) in stress granules is abnormal, resulting in a more than 50% decrease in the K63-linked ubiquitination level of G3BP1. Ultimately, this leads to the impairment of stress granule clearance and a three-fold extension of the time required for cells to restore normal translational function [5,6].

### 3.2. Dormancy and Activation of Ribosomes

In addition to regulating stress granules, SERBP1 also plays a key role in the dynamic balance of ribosomes under nutrient deprivation stress. As the mammalian homologue of the yeast Stm1 protein, SERBP1 can help cells reduce translational consumption during energy scarcity by mediating the formation of “dormant 80S ribosomes”, while maintaining the structural integrity of ribosomes [7]. In vitro reconstitution experiments show that when mTORC1 (Mammalian Target of Rapamycin Complex 1) is inhibited due to nutrient deprivation, SERBP1 can bind to the mRNA entry channel of the 40S ribosomal subunit through its C-terminal α-helical structure, and at the same time interact with the L13a protein of the 60S subunit, stabilizing free 40S/60S subunits into non-translational dormant 80S ribosomes. This process can reduce the overall translational activity of cells by approximately 60%, but the ribosome degradation rate decreases by more than 70% [7,26].

When nutrient conditions are restored, mTORC1 can directly phosphorylate the Ser235 site of SERBP1, disrupting its interaction with ribosomal subunits. Phosphorylated SERBP1 undergoes a conformational change, exposing its RG/RGG domain and binding to 14-3-3 proteins, which are then transported to the cytoplasmic matrix, thereby releasing free 40S/60S subunits to re-participate in translation [7]. Further studies have found that the ribosome dormancy mediated by SERBP1 has the characteristic of “reversible protection”: dormant 80S ribosomes can avoid being recognized and degraded by the autophagic pathway through the interaction between SERBP1 and EIF2α. However, the Ser235Ala mutation of SERBP1 leads to the continuous locking of the ribosome in a dormant state, and even when nutrients are restored, translation cannot be reactivated, ultimately resulting in the stagnation of cell proliferation [26,31]. In mouse embryonic fibroblasts (MEFs), the survival ability of SERBP1-knockout cells after starvation stress is significantly reduced, with a ribosome degradation rate 2.3 times that of wild-type cells, and the protein synthesis rate after nutrient restoration is only 35% of that of wild-type cells, confirming the core role of SERBP1 in ribosome stress protection [7].

### 3.3. Role in Ribosome Biogenesis

In addition to regulating ribosome dormancy and activation, SERBP1 also participates in ribosome biogenesis. It can interact with rRNA (ribosomal RNA) and ribosomal proteins (such as 40S subunit protein RPS2 and 60S subunit protein RPL13a) to promote the assembly of ribosomal subunits. Specifically, SERBP1 binds to the 5′ external transcribed spacer (5′ ETS) of pre-rRNA, facilitating the cleavage of pre-rRNA by ribonuclease III, which is a key step in the maturation of 18S rRNA. In SERBP1-knockdown cells, the cleavage efficiency of pre-rRNA decreases by approximately 45%, leading to the accumulation of immature pre-rRNA and a reduction in the number of mature 40S ribosomal subunits by 30%.

Furthermore, SERBP1 is involved in the quality control of ribosome biogenesis. It can recognize and bind to misfolded ribosomal proteins, recruiting the chaperone protein Hsp70 to promote the correct folding of ribosomal proteins. The absence of SERBP1 leads to the aggregation of misfolded ribosomal proteins, activating the unfolded protein response (UPR) and inhibiting cell proliferation. These findings indicate that SERBP1 plays a dual role in ribosome biogenesis, including promoting pre-rRNA processing and ensuring the correct folding of ribosomal proteins [5,31].

## 4. The Role and Mechanism of SERBP1 in Tumorigenesis and Development

As shown in Table 4, SERBP1 exerts a tumor-type-dependent pro-cancer effect by regulating specific target mRNAs in different tumors.

### 4.1. Liver Cancer

In liver cancer, the long non-coding RNA (lncRNA) circBACH1 can “sponge” miR-656-3p, relieving its translational inhibition on SERBP1 mRNA, leading to the up-regulation of SERBP1 expression. SERBP1 further binds to the 3′UTR region of SERPINE1 mRNA, extending its half-life (from 4.2 h to 7.8 h). Finally, through the extracellular matrix remodeling mediated by SERPINE1, the invasive ability of liver cancer cells is enhanced [8]. Clinical sample analysis shows that the median survival time of liver cancer patients with high SERBP1 expression (n = 86) is 14.6 months, which is significantly shorter than that of patients with low SERBP1 expression (32.8 months), and the expression level of SERBP1 is positively correlated with tumor microvessel density (MVD) [8].

### 4.2. Glioblastoma (GBM)

In glioblastoma (GBM), SERBP1 promotes malignant progression by regulating tumor metabolic reprogramming. Studies have found that SERBP1 can bind to the mRNA of methionine synthase (MTR) to improve its translation efficiency, resulting in an increase in intracellular methionine levels. As a methyl donor, methionine can enhance the histone methylation modification of H3K27me3, inhibit the expression of neurodifferentiation-related genes (such as *NEUROD1* and *MAP2*), and maintain the stemness characteristics of GBM stem cells [3]. The sphere-forming ability of SERBP1-knockout GBM cells decreases by 60%, the sensitivity to temozolomide (TMZ) increases by 2.5 times, and the tumor growth rate in nude mouse xenograft models decreases by 58%, confirming that SERBP1 is a key regulatory factor for the malignant phenotype of GBM [3,32].

### 4.3. Breast Cancer

In breast cancer, the high-glucose microenvironment can induce the K124 lactylation modification of RCC2 (Regulator of Chromosome Condensation 2). Lactylated RCC2 binds to SERBP1 through its N-terminal domain, recruiting SERBP1 to the 5′UTR region of MAD2L1 mRNA. SERBP1 can prevent the degradation of MAD2L1 mRNA (the degradation rate decreases by 40%), leading to the up-regulation of MAD2L1 protein expression, which in turn activates the cell cycle checkpoint and promotes the rapid proliferation of breast cancer cells [9]. Co-immunoprecipitation experiments confirm that the K124A mutation of RCC2 (simulating de-lactylation) significantly weakens its interaction with SERBP1, while the K124L mutation (simulating continuous lactylation) enhances the binding ability, further verifying the key role of lactylation modification in the regulation of SERBP1 [9].

### 4.4. SERBP1 and Cell Cycle Regulation

SERBP1 plays an important role in cell cycle regulation through multiple mechanisms. In addition to regulating MAD2L1 mRNA stability in breast cancer (as mentioned in Section 4.3), it can also interact with Cyclin B1 mRNA in oocytes and tumor cells. Specifically, SERBP1 binds to the 3′UTR of Cyclin B1 mRNA, improving its translation efficiency. Cyclin B1 is a key regulatory protein of the G2/M phase transition. The up-regulation of Cyclin B1 expression promotes the activation of cyclin-dependent kinase 1 (CDK1), thereby promoting cell entry into the mitotic phase. In SERBP1-knockdown cells, the expression level of Cyclin B1 decreases by 58%, leading to G2/M phase arrest and a significant reduction in cell proliferation rate [4,26].

In addition, PKCε-mediated phosphorylation of SERBP1 affects cell cycle progression by regulating ribosome function. Phosphorylated SERBP1 enhances the binding of ribosomes to cell cycle-related mRNAs (such as Cyclin D1 and CDK2 mRNAs), promoting their translation. The inhibition of SERBP1 phosphorylation leads to a decrease in the translation efficiency of these mRNAs, resulting in G1 phase arrest. These findings suggest that SERBP1 regulates cell cycle progression through both mRNA stability regulation and translational regulation [22,26].

### 4.5. Special Cases of Inhibiting Tumor Progression

Although SERBP1 exhibits a tumor-promoting effect in most tumors, it also has a tumor-suppressive function in some tumor types. In triple-negative breast cancer (TNBC), SERBP1 inhibits tumor progression by regulating alternative splicing of genes: SERBP1 interacts with the splicing factor SF3B1 to promote the exon 3 skipping splicing of the ME3 (Methylmalonyl-CoA Epimerase) gene. The resulting truncated ME3 protein (ΔME3) can inhibit the activity of mitochondrial respiratory chain complex I, reduce the production of reactive oxygen species (ROS), and thereby inhibit the invasion of TNBC cells [10]. Clinical data show that the recurrence risk of TNBC patients with high SERBP1 expression (n = 72) decreases by 42%, and the expression level of ΔME3 protein is positively correlated with SERBP1 [10].

In HeLa cells, overexpression of SERBP1 can induce cell apoptosis by activating the p53 pathway. RNA-seq analysis shows that overexpression of SERBP1 leads to differential expression of 89 apoptosis-related genes, among which the expression of pro-proliferation genes such as FOS and FOSB is down-regulated by more than 50%, while the expression of pro-apoptotic genes such as BAX and CASP3 is up-regulated by 2–3 times [10]. Further studies confirm that SERBP1 can interact with the DNA-binding domain of p53 to enhance the transcriptional activation ability of p53 on target genes, while the deletion of SERBP1 leads to a decrease in the acetylation level of p53, making it unable to effectively initiate the apoptotic program [10].

### 4.6. Potential as a Tumor Diagnostic Marker and Therapeutic Target

The abnormal expression of SERBP1 makes it a potential diagnostic marker for various tumors. In ovarian cancer, immunohistochemical analysis shows that the positive expression rate of SERBP1 in tumor tissues is 78.3% (n = 120), which is significantly higher than that in normal ovarian tissues (12.5%, n = 32). Here, “positive expression rate” refers to the percentage of cells with positive SERBP1 protein expression (staining intensity ≥ moderate, positive cell ratio ≥ 10%) among the total detected cells, with the baseline being the SERBP1 expression level in normal ovarian tissues, and the detection standard refers to the immunohistochemical scoring method in reference [33]. High SERBP1 expression is significantly correlated with FIGO stage III-IV and lymph node metastasis [33]. Receiver Operating Characteristic (ROC) curve analysis shows that the area under the curve (AUC) of SERBP1 for diagnosing ovarian cancer is 0.86 (95% CI: 0.80–0.92), with a sensitivity of 81.7% and a specificity of 78.1% [34].

In terms of therapeutic applications, intervention strategies targeting SERBP1 have shown anti-tumor effects in in vitro experiments. In the cisplatin-resistant lung adenocarcinoma cell line (A549/DDP), the expression level of SERBP1 is 2.8 times that of the sensitive cell line (A549). After knocking down SERBP1 by siRNA, the IC50 of cells to cisplatin decreases from 18.6 μmol/L to 7.2 μmol/L, and the apoptosis rate increases from 12.3% to 38.7% [28]. Mechanism studies show that SERBP1 can bind to BRCA1 mRNA to improve its stability (half-life extended from 3.5 h to 6.8 h) and enhance the homologous recombination repair ability. The knockdown of SERBP1 leads to a decrease in BRCA1 expression, reduces DNA damage repair, and reverses cisplatin resistance [28]. In addition, small molecule compound screening finds that the curcumin derivative CUR-197 can bind to the RG/RGG domain of SERBP1 (KD = 2.3 μmol/L), inhibiting its interaction with RNA. It can reduce the tumor volume by 62% in nude mouse xenograft models without obvious toxic reactions [32].

## 5. The Regulatory Role of SERBP1 in the Reproductive System

### 5.1. Heat Stress Protection of Male Germ Cells

SERBP1 has a key stress protection function in the male reproductive system, especially crucial for the survival of germ cells under testicular heat stress conditions. In mouse testes, SERBP1 is mainly expressed in spermatogonia and primary spermatocytes. Heat stress (treatment at 43 °C for 30 min) can induce a 2.5-fold up-regulation of SERBP1 expression, and its expression peak is negatively correlated with the trough of germ cell apoptosis [6]. Mechanism studies show that SERBP1 can recruit the 26S proteasome to the protein aggregation region damaged by heat stress, accelerating the degradation of misfolded proteins. After heat stress, the accumulation of ubiquitinated proteins in the testicular tissue of SERBP1-knockout mice is 3.1 times that of wild-type mice, the apoptosis rate of spermatogonia increases by 4.2 times, and the sperm malformation rate increases from 12.7% to 38.9% [6].

Further studies have found that SERBP1 can also protect germ cells by regulating the dynamic balance of stress granules. Under heat stress conditions, SERBP1 and G3BP1 form stress granules in the cytoplasm of spermatocytes, temporarily storing germ cell-specific mRNAs (such as SYCP3 and PIWIL1) to prevent their degradation. During stress recovery, the ubiquitination of G3BP1 mediated by SERBP1 can quickly clear stress granules, allowing mRNAs to re-enter the translation process. The knockout of SERBP1 leads to a 30–50% reduction in the half-life of these mRNAs, affecting the process of spermatogenesis [6].

### 5.2. Regulation of Female Oocyte Development and Maturation

SERBP1 plays an indispensable role in the development and maturation of oocytes in the female reproductive system. Conditional knockout experiments show that after specific deletion of Serbp1 in oocytes mediated by Zp3-Cre or Gdf9-Cre, the number of preantral follicles and antral follicles in mouse ovaries decreases by 65% and 72% respectively, and oocytes exhibit spindle assembly defects (abnormality rate increased from 8.3% to 45.6%) in the first meiotic division, unable to complete asymmetric division [4]. Further analysis shows that SERBP1 can bind to Cyclin B1 mRNA in oocytes to improve its translation efficiency. The knockout of SERBP1 leads to a 58% decrease in Cyclin B1 protein expression, making oocytes unable to reach the Cyclin B1 threshold required for meiotic maturation, and ultimately arresting at the germinal vesicle breakdown (GVBD) stage [4].

SERBP1 also affects the oocyte microenvironment by regulating the function of granulosa cells. The apoptosis rate of granulosa cells in SERBP1-knockout mice increases by 3.8 times, and the phosphorylation level of Erk1/2 is up-regulated by 2.3 times. The overactivation of Erk1/2 promotes granulosa cells to secrete the pro-apoptotic factor BAX, further aggravating the developmental disorder of oocytes [4]. In addition, the estrous cycle of SERBP1-knockout mice is significantly prolonged (from 4.5 days to 7.2 days), and the serum estrogen level decreases by 40%, confirming the role of SERBP1 in the regulation of female reproductive endocrine [4].

### 5.3. Endometrial Decidualization and Maintenance of Luteal Function

In the endometrium, SERBP1 participates in the progesterone-mediated decidualization process by interacting with Progesterone Receptor Membrane Component 1 (PGRMC1). In human endometrial stromal cells (hESCs), progesterone treatment can induce a 2.1-fold up-regulation of SERBP1 expression. SERBP1 further forms a complex with PGRMC1, activating the Akt signaling pathway. The increased phosphorylation level of Akt promotes the nuclear export of Forkhead Box Protein O1 (FOXO1), relieving its inhibition on decidualization marker genes (such as PRL and IGFBP1) and promoting the differentiation of hESCs into decidual cells [11]. The knockdown of SERBP1 leads to a 60% decrease in the decidualization rate of hESCs, and in the mouse embryo implantation model, the implantation success rate of SERBP1-knockout mice decreases from 78.3% to 23.5% [11].

In luteal tissue, SERBP1 regulates the synthesis and secretion of progesterone. In bovine luteal cells, SERBP1 can bind to the mRNA of Steroidogenic Acute Regulatory Protein (STAR) to enhance its translation efficiency. The increased expression of STAR protein promotes the transport of cholesterol to mitochondria, increasing the progesterone synthesis rate [12]. During the luteal regression stage, the expression of SERBP1 is down-regulated by 55%, leading to a decrease in STAR protein and progesterone secretion, preparing for luteolysis [12]. In addition, SERBP1 can also interact with PGRMC2 to inhibit the apoptosis of luteal cells. The knockout of SERBP1 leads to a 4.2-fold increase in the apoptosis rate of luteal cells, accelerating luteal regression [12].

## 6. The Function of SERBP1 in the Nervous System

### 6.1. Neuronal Chloride Homeostasis and Regulation of GABAergic Transmission

SERBP1 affects neurotransmitter transmission in neurons by regulating the miR-92/KCC2 axis. In rat hippocampal neurons, SERBP1 can form a complex with Ago2 (Argonaute 2) and miR-92, and jointly bind to the 3′UTR region of KCC2 mRNA. The presence of SERBP1 can enhance the inhibitory effect of miR-92 on KCC2 translation, maintaining the KCC2 protein expression at a low level [13]. As a neuronal chloride efflux transporter, the expression level of KCC2 directly affects the intracellular chloride concentration: the knockout of SERBP1 leads to an increase in KCC2 expression, and the intracellular chloride concentration in neurons decreases from 40 mmol/L to 22 mmol/L, converting GABA (γ-Aminobutyric Acid) from an inhibitory neurotransmitter to an excitatory neurotransmitter and enhancing the excitability of hippocampal neurons [13].

Electrophysiological experiments confirm that the amplitude of depolarizing current generated by hippocampal CA1 pyramidal neurons in SERBP1-knockout mice under GABA stimulation is 2.7 times that of wild-type mice, and the induction efficiency of long-term potentiation (LTP) increases by 50%, indicating that SERBP1 affects the synaptic plasticity of neurons by regulating KCC2 [13]. In addition, the expression of SERBP1 is down-regulated by 38% in epilepsy model mice, and restoring SERBP1 expression can reduce the frequency of epileptic seizures by decreasing the KCC2 level, confirming the potential therapeutic value of SERBP1 in nervous system diseases [13].

### 6.2. Pathological Role in Alzheimer’s Disease

SERBP1 is closely related to the pathological process of Alzheimer’s Disease (AD). Its expression level in the brain tissue of AD patients is 1.8 times that of healthy controls, and it is positively correlated with the phosphorylation level of tau protein [5]. Co-immunoprecipitation experiments show that SERBP1 can bind to hyperphosphorylated tau protein (p-tau) and promote the aggregation of tau protein. In vitro experiments, SERBP1 can accelerate the aggregation rate of tau protein, and the formed tau fibers have significantly enhanced toxicity (increasing the rate of neuron apoptosis) [5].

SERBP1 also participates in the pathological transformation of stress granules in the progression of AD. In the neurons of AD model mice (5 × FAD), the stress granule clearance function mediated by SERBP1 is impaired, leading to the co-localization of stress granules with p-tau and Aβ (β-Amyloid Protein) to form “pathological stress granules”. The half-life of this abnormal complex is extended by more than 5 times, which can further recruit other toxic proteins (such as TDP-43) and aggravate neuron damage [5]. The knockdown of SERBP1 can reduce the number of pathological stress granules in the brain tissue of 5 × FAD mice, while decreasing the tau phosphorylation level (the p-tau/tau ratio decreases by 45%) and improving the cognitive function of mice (shortening the latency in the Morris water maze) [5].

### 6.3. Regulatory Mechanism of Neuropathic Pain

In the neuropathic pain model, the expression of SERBP1 is up-regulated in glutamatergic neurons of the primary somatosensory cortex (S1), and its expression level is positively correlated with the pain behavior score [14]. Mechanism studies show that SERBP1 can form a complex with PCIF1 (Phosphorylated CTD Interacting Factor 1), mediating the m6Am modification of Maf1 mRNA. The m6Am modification leads to a 60% decrease in the translation efficiency of Maf1 mRNA, resulting in a reduction in Maf1 protein expression, which in turn relieves its inhibition on RNA polymerase III and promotes tRNA synthesis [14]. The excessive synthesis of tRNA enhances the protein translation rate of neurons, increases the expression of pain-related ion channels (such as Nav1.7), and ultimately exacerbates neuropathic pain [14].

Animal experiments confirm that after specific knockdown of SERBP1 in S1 cortical glutamatergic neurons mediated by AAV (Adeno-Associated Virus), the mechanical pain threshold of model mice increases, the thermal pain latency prolongs, and the anxiety-like behavior (time spent in the central area of the open field test) is improved [14]. In addition, the inhibitor (Cpd-7) of the SERBP1/PCIF1 complex can restore the expression of Maf1 protein by blocking the m6Am modification, showing a good analgesic effect in the neuropathic pain model [14].

## 7. The Role of SERBP1 in Viral Infection

As summarized in Table 5, SERBP1 regulates viral translation or transcription processes by interacting with different viral factors.

### 7.1. Regulation of Dengue Virus (DENV) Replication

SERBP1 is a host-dependent factor necessary for DENV replication. Its expression is up-regulated in DENV-infected HepG2 cells and co-localized with the viral replication complex [15]. Mechanism studies show that the non-structural protein 4A (NS4A) of DENV can bind to the N-terminal domain of SERBP1, recruiting SERBP1 and RACK1 (Receptor for Activated C Kinase 1) to form a complex. This complex can bind to the 5′UTR region of the DENV genome, connecting the viral RNA with the 40S ribosome and promoting the translation of viral RNA [15]. The knockdown of SERBP1 leads to a 3-order-of-magnitude decrease in the DENV viral titer and a 70% reduction in the translation rate of viral RNA, confirming the key role of SERBP1 in DENV translation [15].

Further studies have found that SERBP1 can also promote DENV replication by regulating the stress response of host cells. DENV infection can induce the formation of stress granules mediated by SERBP1. These stress granules can temporarily store host antiviral-related mRNAs (such as IFN-β and ISG15), inhibiting the host antiviral immune response. The knockout of SERBP1 leads to the inability to form stress granules effectively, and the expression of host antiviral genes is up-regulated by 2–3 times, restricting the replication of DENV [15].

### 7.2. Cellular Transformation by Kaposi’s Sarcoma-Associated Herpesvirus (KSHV)

The viral interleukin-6 (vIL-6) encoded by KSHV can promote cellular transformation by regulating the deacetylation modification of SERBP1. In KSHV-infected human umbilical vein endothelial cells (HUVECs), vIL-6 can activate SIRT3 (Sirtuin 3), leading to the deacetylation of SERBP1 at the K213 site. Deacetylated SERBP1 has a 60% decrease in its binding ability to Lipt2 (Lipoyltransferase 2) mRNA, resulting in an accelerated degradation rate of Lipt2 mRNA and a reduction in Lipt2 protein expression [16]. As a key regulatory factor of ferroptosis, the decreased expression of Lipt2 inhibits ferroptosis (the level of lipid peroxidation decreases by 55%), allowing KSHV-infected cells to avoid ferroptosis and achieve malignant transformation [16].

The SIRT3-specific inhibitor 3-TYP can restore the expression level of Lipt2 by inhibiting the deacetylation of SERBP1, inducing ferroptosis in KSHV-transformed cells. In the nude mouse model of KSHV-related sarcoma, 3-TYP can reduce the tumor volume [16]. In addition, the K213Q mutation of SERBP1 (simulating acetylation) significantly weakens its ability to promote KSHV transformation, further verifying the role of acetylation modification in the regulation of SERBP1 function [16].

### 7.3. Transcriptional Activation of Avian Leukosis Virus (ALV-K)

SERBP1 can enhance the transcriptional activity of the virus by interacting with the long terminal repeat (LTR) of ALV-K. The LTR region of ALV-K contains an 11 nt SERBP1-binding site (5′-GTGGTATGATC-3′). SERBP1 can bind to this site through its RG/RGG domain, recruiting the transcriptional co-activator p300, increasing the acetylation level of histone H3K27 in the LTR region by 2.3 times, and promoting the transcription of the viral genome [17]. In SERBP1-knockdown DF-1 cells (chicken fibroblasts), the viral titer of ALV-K decreases by 2 orders of magnitude, and the transcription rate of viral RNA decreases by 65% [17].

Sequence analysis shows that this 11 nt binding site is absent in other ALV subtypes (such as ALV-A and ALV-B), resulting in the inability of SERBP1 to promote the transcription of these subtypes, explaining the unique host adaptability of ALV-K [17]. In addition, small molecule inhibitors (ALV-K-In-1) targeting the binding of SERBP1 to LTR can inhibit the replication of ALV-K in chicken models, providing a new strategy for the prevention and control of avian leukosis [17].

## 8. The Role of SERBP1 in Other Physiological and Pathological Processes

### 8.1. Regulation of Metabolism-Related Diseases

In the diabetic lower limb ischemia model, Adipsin (Complement Factor D) can improve angiogenesis by interacting with SERBP1. The expression of Adipsin in the ischemic lower limb muscles of diabetic mice is down-regulated by 45%, while exogenous Adipsin can bind to SERBP1, disrupting the interaction between SERBP1 and SERPINE1 mRNA. The half-life of SERPINE1 mRNA is shortened from 6.8 h to 3.2 h, and the expression of SERPINE1 protein is reduced, relieving its inhibition on the VEGFR2 (Vascular Endothelial Growth Factor Receptor 2) signaling pathway [35]. Adipsin treatment can increase the capillary density of the lower limbs of diabetic mice by 2.1 times and improve the blood perfusion recovery rate by 65%. The knockout of SERBP1 completely blocks the therapeutic effect of Adipsin, confirming that SERBP1 is a key downstream factor of Adipsin-mediated angiogenesis [35].

In non-alcoholic steatohepatitis (NASH), SERBP1 promotes disease progression by regulating lipid metabolism. The expression of SERBP1 in the liver tissue of NASH patients is up-regulated by 2.4 times. SERBP1 can bind to the mRNA of SREBP1 (Sterol Regulatory Element-Binding Protein 1) to improve its translation efficiency. The increased expression of SREBP1 promotes the expression of fatty acid synthase (FASN) and acetyl-CoA carboxylase (ACC), leading to an increase in triglyceride accumulation in hepatocytes [36]. In the SERBP1-knockdown NASH mouse model, the degree of hepatic steatosis is reduced by 60%, and the expression of inflammatory factors (TNF-α and IL-6) is decreased by 45%, confirming the pro-pathological role of SERBP1 in NASH [36].

### 8.2. Mechanism of Association with Cardiovascular Diseases

Single Nucleotide Polymorphisms (SNPs) of SERBP1 are associated with the risk of ischemic heart disease. Among 2164 subjects (836 ischemic heart disease patients and 1328 controls), carriers of the C allele at the rs1058074 locus of SERBP1 have a significantly reduced risk of disease (OR = 0.63, 95% CI: 0.43–0.93, *p* = 0.02), and this allele is associated with a decrease in serum low-density lipoprotein cholesterol (LDL-C) levels [37]. Mechanism studies show that the rs1058074 locus is located in the promoter region of SERBP1, and the C allele reduces the binding ability of the transcription factor SP1, leading to a 35% down-regulation of SERBP1 expression. The decreased expression of SERBP1 inhibits the cholesterol synthesis mediated by SREBP1, reducing the production of LDL-C [37].

In the atherosclerosis model, oxidized low-density lipoprotein (oxLDL) can up-regulate the expression of SERBP1 by activating the NLRP3 inflammasome. In oxLDL-treated macrophages, the activation of NLRP3 promotes the cleavage of Caspase-1, thereby inducing the expression of SERBP1. This leads to SERBP1 further binding to IL-1β mRNA, enhancing its translation efficiency, exacerbating the inflammatory response, and promoting the formation of atherosclerotic plaques [38]. The NLRP3 inhibitor MCC950 can reduce the secretion of IL-1β by inhibiting the expression of SERBP1, reducing the area of atherosclerotic plaques in ApoE^−/−^ mice [38].

## 9. Research Prospects

Although significant progress has been made in the research of SERBP1, there are still several key issues to be solved in its molecular mechanism and clinical application. At the molecular mechanism level, as an intrinsically disordered protein, the structural basis of SERBP1’s interaction with RNA and proteins has not been fully elucidated. In the future, it is necessary to use cryo-electron microscopy technology to analyze the complex structures of SERBP1 bound to different ligands, clarify the action mode of its RG/RGG domain, and the regulatory mechanism of post-translational modifications (such as methylation and phosphorylation) on the complex structure [39]. In addition, the interactome of SERBP1 varies among different cell types. It is necessary to map the cell-specific SERBP1 interaction network using single-cell proteomics technology to reveal the molecular basis of its functional diversity [5].

In terms of clinical transformation, the diagnostic value of SERBP1 as a tumor marker still needs to be verified by multi-center, large-sample studies. In particular, it is necessary to develop minimally invasive detection methods based on SERBP1 (such as serum exosome SERBP1 detection) to improve the convenience and sensitivity of diagnosis [40]. In therapeutic applications, targeted drugs for SERBP1 (such as small molecule inhibitors and siRNA) need to further optimize the administration method (such as targeted drug delivery systems) to reduce off-target effects, and conduct early clinical trials to evaluate their safety and effectiveness [16,32]. In addition, the therapeutic potential of SERBP1 in neurodegenerative diseases and metabolic diseases also requires more animal experiments and preclinical studies to provide new strategies for the treatment of related diseases [14,35].

In the field of cross-species research, current functional studies of SERBP1 mainly focus on humans, mice, and a few model organisms, while its mechanism of action in agricultural animals (such as cattle and chickens) is still unclear. In the future, it is necessary to carry out research on SERBP1 in agricultural animal diseases (such as avian leukosis and bovine luteal dysfunction) to provide new targets for the prevention and control of livestock and poultry diseases [12,17]. At the same time, the evolutionary conservation analysis of SERBP1 can reveal the evolutionary law of its function, providing clues for understanding the conserved mechanism of life activities [41].

In conclusion, as a multifunctional RNA-binding protein, SERBP1 plays a key role in various fields such as cellular stress, tumors, reproduction, nerves, and viral infection. With the advancement of research techniques, the molecular mechanism of SERBP1 will be further clarified, and its application potential in disease diagnosis and treatment will be gradually realized, providing important support for human health and the prevention and control of animal diseases.

## 10. Conclusions

SERBP1 is a versatile RNA-binding protein that orchestrates multiple cellular processes. Its structural flexibility, driven by RG/RGG domains and post-translational modifications, allows it to interact with diverse RNAs and proteins, influencing stress responses, ribosome dynamics, tumor progression, reproductive biology, neural function, and viral replication. Recent insights into its interaction with PARP1, role in ribosome biogenesis, and cell cycle regulation further expand our understanding of its functional complexity. Future research should focus on elucidating its structural basis for molecular interactions, exploring cell-type-specific mechanisms, and translating these findings into diagnostic and therapeutic strategies for related diseases.

## Figures and Tables

**Figure 1 cells-14-01705-f001:**
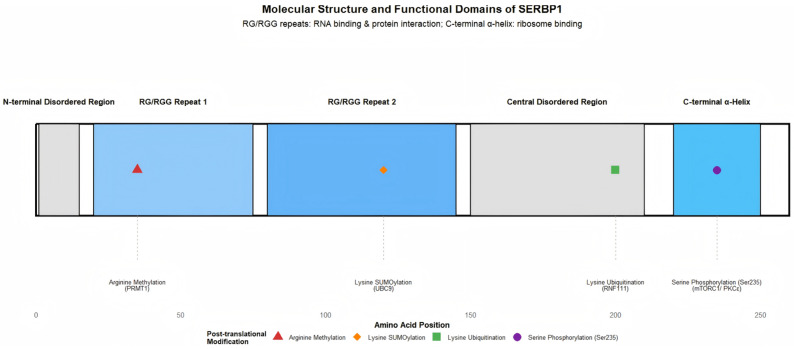
Molecular structure and functional domains of SERBP1. Note: This figure labels the core domains of SERBP1, including the N-terminal disordered region, two RG/RGG repeat sequences (core regions for RNA binding and protein interaction), and the C-terminal α-helical structure (binds to the mRNA entry channel of the 40S ribosomal subunit). It also marks key post-translational modification sites (Ser235 phosphorylation site, lysine SUMOylation site, and arginine methylation site). The functions of each domain are based on references [13,18,19].

**Figure 2 cells-14-01705-f002:**
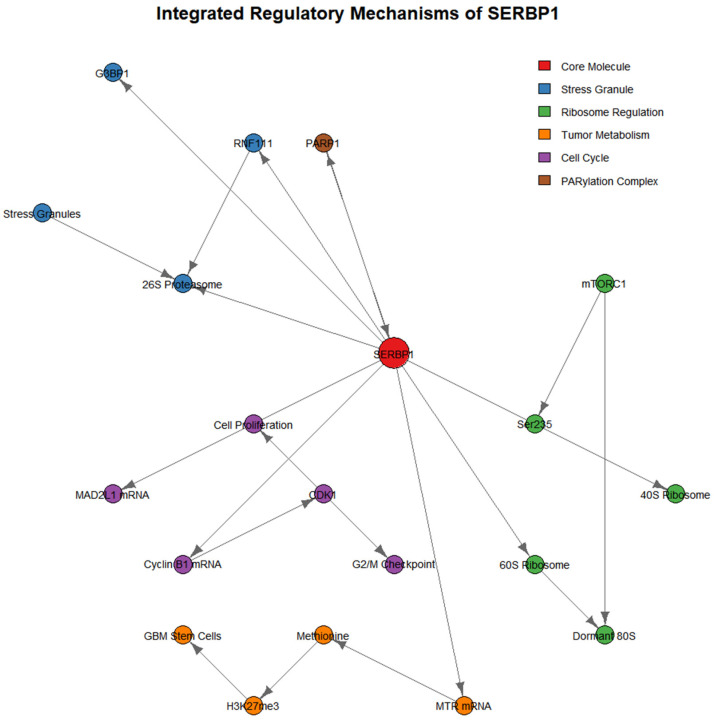
Integrated regulatory mechanisms of SERBP1. Note: This figure integrates SERBP1’s pathways in stress granule assembly/clearance (G3BP1 interaction, RNF111-mediated ubiquitination), ribosome dormancy/activation (mTORC1-regulated Ser235 phosphorylation), tumor metabolic reprogramming (MTR mRNA regulation), and cell cycle regulation (MAD2L1 mRNA regulation), clarifying the core functional molecules and upstream-downstream relationships. References [5,6,7,8,9] are cited.

**Table 1 cells-14-01705-t001:** Core biological functions of SERBP1 and associated cellular processes.

Functional Category	Key Cellular Processes	Biological Outcome	References
Cellular Stress Response	Stress granule (SG) assembly/clearance; ribosome dormancy/activation	Protect cells from stress-induced damage; maintain ribosome integrity during nutrient scarcity	[6,7]
Tumorigenesis	mRNA stability regulation; metabolic reprogramming; alternative splicing	Promote/inhibit tumor progression (tumor-type dependent); affect drug sensitivity	[3,8,9,10]
Reproductive Regulation	Germ cell stress protection; oocyte maturation; endometrial decidualization; luteal function	Maintain spermatogenesis; support follicle development; ensure embryo implantation	[4,6,11,12]
Nervous System Function	Chloride homeostasis; tau aggregation; neuropathic pain signaling	Regulate GABAergic transmission; modulate AD pathology; control pain sensitivity	[5,13,14]
Viral Infection	Viral RNA translation; host immune response; viral transcription	Facilitate replication of DENV, KSHV, ALV-K	[15,16,17]

**Table 2 cells-14-01705-t002:** Key post-translational modifications of SERBP1 and their functional effects.

Modification Type	Mediating Enzyme/Target Site	Subcellular Localization Change	Functional Outcome
Arginine Methylation	PRMT1 (RG/RGG domain)	Hypomethylation: Nucleus/nucleolus; Normal methylation: Cytoplasm/SGs	Regulates RNA/protein interaction; promotes SG recruitment [20,21]
Phosphorylation	PKCε; mTORC1 (Ser235)	Cytoplasmic matrix (after mTORC1-mediated phosphorylation)	Modulates ribosome function (PKCε); releases dormant ribosomes (mTORC1) [7,22]
SUMOylation	Lysine residues (multiple sites)	Nuclear translocation (under oxidative stress)	Alters PML-NB formation; enhances p53-mediated apoptosis [23,24]
Ubiquitination	Ubiquitin-proteasome system	Cytoplasm (degradation)	Degrades SERBP1 under long-term stress; prevents toxic SG accumulation [25]

**Table 3 cells-14-01705-t003:** Summary of SERBP1 target RNAs and interacting proteins.

Category	Molecule Name	Binding Region/Interaction Site	Function	References
Target RNA	SERPINE1 mRNA	3′UTR	Extends half-life, promotes liver cancer invasion	[8,28]
Target RNA	MTR mRNA	—	Improves translation efficiency, maintains GBM stem cell characteristics	[3]
Target RNA	MAD2L1 mRNA	5′UTR	Inhibits degradation, activates breast cancer cell cycle checkpoint	[9]
Target RNA	Lipt2 mRNA	—	Regulates degradation, affects KSHV-induced cellular transformation	[16]
Interacting Protein	G3BP1	RG/RGG domain	Promotes stress granule assembly and clearance	[6,29]
Interacting Protein	PRMT1	RG/RGG domain	Mediates arginine methylation, regulates subcellular localization	[20]
Interacting Protein	PARP1	—	Forms PARylation-dependent complex, regulates splicing and cell division	[5]
Interacting Protein	RNF111	—	Promotes K63-linked polyubiquitination of G3BP1	[6]

**Table 4 cells-14-01705-t004:** Tumor-promoting mechanisms of SERBP1 in different cancers.

Tumor Type	Upstream Regulator	Target Molecule (mRNA)	Functional Effect	Clinical Correlation
Liver Cancer	circBACH1 (sponges miR-656-3p)	SERPINE1 (3′UTR)	Extends SERPINE1 half-life (4.2 → 7.8 h); enhances cell invasion [8]	High SERBP1: Median survival 14.6 months (vs. 32.8 months for low SERBP1) [8]
Glioblastoma (GBM)	—	MTR	Increases methionine levels; enhances H3K27me3; maintains stemness [3]	SERBP1 knockout: 60% reduced sphere formation; 2.5× higher TMZ sensitivity [3,32]
Breast Cancer	Lactylated RCC2 (K124)	MAD2L1 (5′UTR)	Reduces MAD2L1 degradation (40%); activates cell cycle checkpoint [9]	High SERBP1: Accelerated proliferation; linked to high-glucose microenvironment [9]

**Table 5 cells-14-01705-t005:** SERBP1’s roles in viral infection.

Virus Type	Viral Factor Interacting with SERBP1	Host/Viral Target	Functional Outcome
DENV	NS4A	Viral 5′UTR; Host antiviral mRNAs	Promotes viral translation; inhibits immune response [15]
KSHV	vIL-6 (indirect via SIRT3)	Lipt2 mRNA	Inhibits ferroptosis; supports cellular transformation [16]
ALV-K	LTR (11 nt binding site)	Viral LTR (H3K27 acetylation)	Enhances viral transcription; mediates subtype-specific replication [17]

## Data Availability

All data supporting the findings of this review are derived from publicly available literature cited within the manuscript. No original experimental data were generated for this review.

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
