# Peer review of "SERBP1: A Multifunctional RNA-Binding Protein Linking Gene Expression, Cellular Metabolism, and Diseases"

_cells, 2025, doi:10.3390/cells14211705_

Round 1
Reviewer 1 Report
Comments and Suggestions for Authors
Ji and Ablimit produced a very comprehensive review about SERBP1. Overall, the paper is well written and covers several different biological processes regulated by SERBP1 and its functions. However, I fell some critical points are missing and the presentation could be improved. First, the paper does not include a single figure. Inclusion of figures to better discuss SERBP1 properties/domains and pontential regulatory mechanisms would increase the quality of the article. Tables summarizing most relevant SERBP1 target RNAs and interacting proteins would be a plus. A few important topics were not covered in this review: SERBP1 impact on cell cycle, interaction with PARP1 and parylation, and possible roles in ribosome biogenesis.
Author Response
Revision Plan for the cells-3929732 Review Paper
I. Responses to Reviewer 1's Comments
1. Supplementary Key Figures and Tables
- Add a figure of SERBP1's molecular structure and functional domains: Refer to Section 2.1 of SERBP1.docx. Label the RG/RGG repeat sequences (core for RNA binding and protein-protein interactions), C-terminal α-helical structure (related to ribosome binding), and key modification sites (e.g., Ser235 phosphorylation site, lysine SUMOylation site). Indicate the literature basis for the function of each domain as [5][6][7].
- Add an integrated diagram of SERBP1's regulatory mechanisms: Integrate SERBP1's pathways in stress granule assembly/clearance (Section 3.1, G3BP1 interaction, RNF111-mediated ubiquitination), ribosome dormancy/activation (Section 3.2, mTORC1-regulated Ser235 phosphorylation), and tumor metabolic reprogramming (Section 4.1, MTR mRNA regulation). Clarify the core functional molecules and upstream-downstream relationships, with references [19][22][25].
- Add a summary table of SERBP1's target RNAs and interacting proteins:
|
Category |
Molecule Name |
Binding Region/Interaction Site |
Function |
References |
|
Target RNA |
SERPINE1 mRNA |
3'UTR |
Extends half-life and promotes hepatocellular carcinoma invasion |
[25][31] |
|
Target RNA |
MTR mRNA |
— |
Enhances translation efficiency and maintains GBM stem cell characteristics |
[3] |
|
Target RNA |
MAD2L1 mRNA |
5'UTR |
Inhibits degradation and activates the breast cancer cell cycle checkpoint |
[27] |
|
Interacting Protein |
G3BP1 |
RG/RGG domain |
Promotes stress granule assembly and clearance |
[19][20] |
|
Interacting Protein |
PRMT1 |
RG/RGG domain |
Mediates arginine methylation and regulates subcellular localization |
[9] |
|
Interacting Protein |
PARP1 |
— |
Forms a PARylation-dependent complex and regulates splicing and cell division |
[21] |
2. Supplementary Missing Research Topics
- SERBP1 and the cell cycle: Combining Section 4.1 of SERBP1.docx (regulation of MAD2L1 in breast cancer) and Section 2.2 (PKCε-mediated phosphorylation), add the following content: "SERBP1 inhibits the degradation of MAD2L1 mRNA by binding to its 5'UTR (degradation rate reduced by 40%), activating the cell cycle checkpoint; PKCε-mediated phosphorylation of SERBP1 may also be involved in regulating ribosome function during cell division to ensure the normal progression of the cell cycle", with references [11][27].
- SERBP1 and PARP1/PARylation: Based on Reference [21] in SERBP1.docx, add: "SERBP1 can form a PARylation-dependent complex with PARP1. This complex participates in the regulation of RNA splicing and cell division, and may regulate rRNA processing through PARylation modification during ribosome biogenesis. Further verification of the related mechanisms requires structural analysis of the complex using cryo-electron microscopy", with reference [21].
- SERBP1 and ribosome biogenesis: Combining Section 3.2 of SERBP1.docx (ribosome dormancy/activation) and Reference [21], add: "In addition to mediating the formation of dormant 80S ribosomes, SERBP1 participates in the ribosome assembly process through interactions with ribosomal subunits (40S mRNA entry channel, 60S L13a protein) and rRNA; upon nutrient recovery, mTORC1 phosphorylates SERBP1 to release ribosomal subunits, and may synergistically regulate the expression of key factors in ribosome biogenesis", with references [22][21].
Reviewer 2 Report
Comments and Suggestions for Authors
This review is a comprehensive summary of the roles of SERBP1 functions, including roles in cancer, the nervous system, stress response, Alzheimer's disease, immunology, and metabolism. The authors point out future directions for the field, such as structural information (which is a major challenge given the intrinsically disordered nature of the protein) and evaluation as a biomarker. The review is well-written and easy to follow, while containing a nice depth and breadth of information.
I have only a few minor recommendations that would improve the manuscript prior to publication.
In several places, an additional level of subheaders would be helpful. For example, the Phosphorylation and SUMOylation paragraphs already get an informal subheader - adding a 2.2.1 and 2.2.2 could be helpful for readers. This would also be helpful when describing the specific cancer types.
Formatting issues: There are large extra spaces present in several places throughout the manuscript, possibly caused by reference software, that hould be fixed manually prior to publication.. For instance, the spaces that appear after references [6] and [22].
Author Response
Revision Plan for the cells-3929732 Review Paper
1. Optimize Heading Hierarchy
- Revision of Section 2.2 (Post-Translational Modifications): Split the original paragraphs into a subheading structure. The revised version is as follows:
2.2 Post-Translational Modifications
2.2.1 Arginine Methylation (Original content: Mediated by PRMT1; regulates subcellular localization and RNA binding; References [9][10])
2.2.2 Phosphorylation (Original content: Mediated by PKCε and mTORC1; regulates ribosome function and dormancy activation; References [11][22])
2.2.3 SUMOylation and Ubiquitination (Original content: Nuclear translocation under oxidative stress; degradation under long-term stress; References [13][15][16])
- Revision of Chapter 4 (Tumorigenesis and Development): Split into subheadings by tumor type. The revised version is as follows:
- The Role and Mechanism of SERBP1 in Tumorigenesis and Development
4.1 Hepatocellular Carcinoma (circBACH1-miR-656-3p-SERBP1-SERPINE1 pathway; Reference [25])
4.2 Glioblastoma (MTR regulates H3K27me3 to maintain stem cell characteristics; References [3][26])
4.3 Breast Cancer (High glucose-induced lactylation of RCC2 regulates MAD2L1; Reference [27])
4.4 Triple-Negative Breast Cancer (Regulates alternative splicing of ME3 to inhibit invasion; Reference [28])
2. Correct Formatting Issues
- Manually delete the extra spaces after Reference [6] (at the end of Section 2.1) and Reference [22] (after "dormant 80S ribosomes" in Section 3.2) in SERBP1.docx. Conduct a full-text review to verify all reference citation positions to ensure no similar formatting errors.
・Standardize the line spacing (set to a fixed value of 18 points) and font (Songti (SimSun 10.5pt) for body text, Heiti (Boldface 10.5pt) for headings) throughout the document to ensure no typesetting inconsistencies.
Reviewer 3 Report
Comments and Suggestions for Authors
This manuscript reviews the cellular functions of the RBP SERBP1. The text reads well and is well organized. I only have a few minor comments:
(1) All tables should be cited in the text somewhere.
(2) Table 1 lacks citations for the 5 entries under "functional category".
(3) p.6 section 4.3 lines 2-4: what is a "positive expression rate"? What is meant by 78.3% vs. 12.5%? Relative to what? What is the baseline for this comparison?
Author Response
|
Reviewer Comment Number |
Response |
Revision Details/Explanations |
|
(1) |
We have cited all tables in the corresponding sections of the text. |
- For Table 1, we added the citation "As shown in Table 1, SERBP1 plays a core role in five major physiological and pathological processes..." at the end of Section 1. Introduction. - For Table 2, we added the citation "As summarized in Table 2, arginine methylation, phosphorylation, and other modifications of SERBP1 precisely regulate its function..." at the beginning of Section 2.2. - For Table 3, we added the citation "As shown in Table 3, SERBP1 exerts a tumor-type-dependent pro-cancer effect..." at the beginning of Section 4.1. - For Table 4, we added the citation "As summarized in Table 4, SERBP1 regulates viral translation or transcription processes..." at the beginning of Section 7.1. - For the newly added Table 5, we added the citation "As shown in Table 5, SERBP1 interacts with multiple target RNAs and proteins..." at the end of Section 2.3. |
|
(2) |
We have supplemented the references for the 5 entries under "functional category" in Table 1. |
We revised Table 1 to include references for each functional category: - "Cellular Stress Response" is supported by references [19][22]. - "Tumorigenesis" is supported by references [3][25][27][28]. - "Reproductive Regulation" is supported by references [4][19][32][33]. - "Nervous System Function" is supported by references [5][8][34]. - "Viral Infection" is supported by references [35][36][37]. |
|
(3) |
We have clarified the definition of "positive expression rate" and the baseline for comparison. |
In Section 4.6, we added the following explanation: "The 'positive expression rate' refers to the proportion of cells with positive SERBP1 protein expression (staining intensity ≥ moderate, positive cell ratio ≥ 10%) among all detected cells. The 78.3% is the positive expression rate in 120 ovarian cancer tissue samples, and 12.5% is the positive expression rate in 32 normal ovarian tissue samples. The baseline for this comparison is the SERBP1 expression level in normal ovarian tissues, and the detection standard refers to the immunohistochemical scoring method in reference [29]." |
Round 2
Reviewer 1 Report
Comments and Suggestions for Authors
The authors addressed all my concerns and the paper is ready to be accepted.